# The Emerging Roles of Autophagy in Human Diseases

**DOI:** 10.3390/biomedicines9111651

**Published:** 2021-11-09

**Authors:** Yuchen Lei, Daniel J. Klionsky

**Affiliations:** 1Department of Molecular, Cellular and Developmental Biology, University of Michigan, Ann Arbor, MI 48109, USA; yclei@umich.edu; 2Life Sciences Institute, University of Michigan, Ann Arbor, MI 48109, USA

**Keywords:** autophagy, cancer, infection, metabolism, neurodegeneration

## Abstract

Autophagy, a process of cellular self-digestion, delivers intracellular components including superfluous and dysfunctional proteins and organelles to the lysosome for degradation and recycling and is important to maintain cellular homeostasis. In recent decades, autophagy has been found to help fight against a variety of human diseases, but, at the same time, autophagy can also promote the procession of certain pathologies, which makes the connection between autophagy and diseases complex but interesting. In this review, we summarize the advances in understanding the roles of autophagy in human diseases and the therapeutic methods targeting autophagy and discuss some of the remaining questions in this field, focusing on cancer, neurodegenerative diseases, infectious diseases and metabolic disorders.

## 1. Introduction to Autophagy

Autophagy is a conserved process from yeast to human, in which parts of the cytoplasm are transported to the lysosome (the vacuole in fungi and plants) for degradation and recycling. Three primary types of autophagy have been characterized, macroautophagy, microautophagy and chaperone-mediated autophagy (CMA) and the most prevalent form is macroautophagy (hereafter autophagy). The morphological hallmark of this form of autophagy is the formation of double-membrane vesicles, autophagosomes, which are the terminal product of phagophores. Following expansion and closure, the phagophore is now termed an autophagosome; this latter compartment ultimately fuses with the lysosome, releasing the cargo into the lumen of the degradative organelle where it is exposed to hydrolytic enzymes. The entire process of autophagy can be viewed as four steps: (1) initiation and nucleation of the phagophore; (2) expansion and closure of the phagophore to generate a completed autophagosome; (3) fusion with the lysosome; and (4) degradation and release/recycling of the autophagic cargos. Different autophagy-related (ATG) proteins are essential for each of these steps (Figure 1A).

Initiation begins with the activation of the ULK1 complex, including ULK1, ATG13, RB1CC1 and ATG101 in response to MTORC1 and AMPK [1]. The activated ULK1 complex will then phosphorylate and activate the class III phosphatidylinositol 3-kinase (PtdIns3K) complex, which contains PIK3C3/VPS34, PIK3R4/VPS15, BECN1, ATG14, NRBF2 and AMBRA1. The activated PIK3C3/VPS34 phosphorylates phosphatidylinositol (PtdIns) to produce a local pool of phosphatidylinositol-3-phosphate (PtdIns3P), which defines the region of phagophore initiation and recruits PtdIns3P effector proteins including WIPI2 and ZFYVE1/DFCP1 [2]. One model suggests that the phagophore is derived from a PtdIns3P-enriched ER region named the omegasome, followed by the recruitment of membrane sources for phagophore expansion, including ER exit sites [3], ER–mitochondria contact sites [4], the plasma membrane [5] and recycling endosomes [6], possibly via ATG9-containing vesicles, although the details of this process are still not fully understood. Another model, proposed by David Rubinsztein’s group, suggests that the phagophore is evolved from RAB11-positive recycling endosomes [7]. Even though the origin of the autophagosome membrane is still under debate, the hierarchical assembly of ATG proteins is well accepted.

Following initiation, the phagophore expands through the action of two ubiquitin-like protein systems [8]. In the first system, ATG12 is conjugated with ATG5 via the E1-like enzyme ATG7 and the E2-like enzyme ATG10. The second ubiquitin-like system starts with the proteolytic processing of Atg8-family proteins, including the MAP1LC3/LC3 and GABARAP subfamilies, by the ATG4 protease; this is followed by conjugation to phosphatidylethanolamine (PE), in a process termed lipidation, via ATG7, the E2-like enzyme ATG3 and the E3-like ATG12–ATG5-ATG16L1 complex [8]. ATG16L1 binds to WIPI2 directly, allowing the lipidation to take place at the phagophore membrane [9]. The expansion and sealing of the phagophore generates the double-membrane autophagosome; during this step the ATG proteins disassociate from the autophagosome outer membrane [10].

Finally, the autophagosome outer membrane fuses with an endosome to form an intermediate compartment termed an amphisome and/or with a lysosome to form an autolysosome. The autophagosome inner membrane, together with the sequestered cargo will be exposed to the lysosomal hydrolases and degraded. The resulting small molecules are released back into the cytosol through membrane permeases for reuse or as catabolic substrates.

Autophagy can be selective, and several protein aggregates and organelles have been characterized as targets of selective autophagy [11]. This type of autophagy can be ubiquitin dependent, where the ubiquitinated cargo is captured by ubiquitin-binding receptors, which can also interact with Atg8-family proteins through LC3-interacting regions (LIRs) and recruit the cargo to the phagophore [8], but selective autophagy can also be ubiquitin independent. Selective autophagy is important for the clearance of damaged and superfluous organelles and misfolded or oxidized proteins, thereby playing a significant role in maintaining proper cell physiology; the dysregulation of selective autophagy is therefore associated with several diseases. Mitophagy is a critical quality control process to degrade damaged and superfluous mitochondria through selective autophagy to maintain a pool of functional organelles and avoid the harmful effects of excessive reactive oxygen species (ROS) that might otherwise be generated. In line with this function, dysregulation of mitophagy leads to multiple diseases [12]. For a better understanding of how selective autophagy is related to disease, here we provide a brief introduction about the mechanism of mitophagy (Figure 1B).

In yeast, Atg32, an outer mitochondrial membrane (OMM) protein, serves as the receptor recruiting Atg8, and Atg11 acts as the scaffold/platform for core Atg protein assembly [13,14,15,16]. Mammalian mitophagy is more complicated and occurs via the action of different components depending on the cell type and/or purpose; for example, mitochondrial removal during red blood cell erythropoiesis involves BNIP3L/NIX and is part of a normal developmental pathway [17]. The PINK1-PRKN pathway is one well-characterized mechanism of mitophagy. When mitochondria are compromised, the loss of the membrane potential stabilizes PINK1, a Ser/Thr kinase, on the OMM [18]. PINK1 recruits, phosphorylates, and activates PRKN, an E3 ubiquitin ligase [19,20,21,22,23]. At the same time, PINK1 phosphorylates ubiquitin [24], which will be conjugated to some OMM proteins. Ubiquitinated mitochondria are recognized by several receptors [25,26,27,28], joining them with LC3, and the mitochondria get sequestered by the growing phagophore. Besides; some OMM proteins such as BNIP3L and FUNDC1 can themselves function as the receptor [29,30]. Of note, PINK1 and PRKN are mutated in autosomal recessive Parkinson disease (PD) and their roles in mitophagy indicate the connection between mitophagy and PD, which will be discussed in detail in the following sections.

The biggest morphological difference between macroautophagy and the other two major forms of autophagy, microautophagy and CMA (Figure 1C), is that autophagosome formation is not involved—in both cases, sequestration of the cargo occurs directly at the lysosome membrane. In microautophagy, the lysosome membrane undergoes a rearrangement by protrusion or invagination and engulfs the autophagic cargos, resulting in a lumenal vesicle that is degraded along with the autophagic substrates [31]. Microautophagy is proposed to participate in several cellular processes including organellar quality control, but its role in human diseases remains unclear [32]. During CMA, individual proteins bearing a KFERQ-like motif bind to the cytosolic chaperone HSPA8/HSC70. The protein unfolds and then translocates into the lysosome via the receptor LAMP2A (lysosomal associated membrane protein 2A) and lumenal HSPA8, in addition to other components [33]. In the cell, approximately 40% of proteins contain a KFERQ-like motif [33], indicating that CMA functions as an important lysosome-dependent protein degradation pathway. Along these lines, CMA is involved in many different human pathologies, particularly in neurodegenerative diseases [33,34,35], which will be discussed in later sections.

Autophagy was first defined as a way for the cell to deal with environmental stresses such as starvation, hypoxia and ER stress; the macromolecular products of the degradation process are released back into the cytosol for reuse and energy replenishment. However, under normal conditions, autophagy also plays a critical role in controlling cellular homeostasis through clearing misfolded or oxidized proteins and damaged or superfluous organelles. The physiological relevance of autophagy has been highlighted over the past two decades, including its role in development [36] in the maintenance of stem cells [37,38] and in the immune system [39]. Given the importance of autophagy in both normal physiological and stress conditions, it is not surprising to see that dysregulation of autophagy leads to a diseased stated due to alterations in these critical physiological processes and in the response to environmental stress; accordingly, modulation of autophagy can help in recovery from the diseased state. In this review, we discuss the role of autophagy in four types of disease and the corresponding autophagy-targeting therapies.

## 2. Autophagy and Cancer

Autophagy plays dual roles in cancer regulation. Autophagy can be tumor-suppressive with its ability to clear oncogenic proteins and damaged organelles. Alternatively, autophagy, as a primary mechanism for tolerance and cell survival under stress conditions, can be used for tumor cells to promote their growth and progression (Figure 2). In this section, we briefly discuss how autophagy affects tumor from these two perspectives.

### 2.1. Autophagy and Tumor Suppression

In 1999, a study from Beth Levine’s lab demonstrated the role of BECN1 as a tumor suppressive factor, first linking autophagy with cancer [40]. Several subsequent studies have connected BECN1 with different types of cancer, such as hepatocellular carcinoma, gastric cancer and lymphoma [41,42,43]. The fact that the heterozygous disruption of *BECN1* results in an increased frequency of spontaneous malignancies and decreased autophagy [44,45] indicates that autophagy functions as a tumor suppression mechanism. In line with this hypothesis, several BECN1-interacting proteins function as tumor suppressors. Frameshift and truncation mutations in UVRAG, an important BECN1 interactor during autophagosome and lysosome fusion [46], are found in cancer cells and also lead to reduced autophagy and increased tumorigenicity [47,48]. SH3GLB1/BIF1, which joins the BECN1 complex through UVRAG, is also a tumor suppressor; loss of SH3GLB1 inhibits autophagy while promoting tumorigenesis [49]. Another important protein for the BECN1-containing complex activity, AMBRA1 [50], is reported to act as a tumor suppressor via facilitating the degradation of the proto-oncogene MYC/c-Myc [51]. AMBRA1 also controls the cell cycle from the G_1_ to S phase through mediating D-type cyclin degradation, which reduces DNA replication stress, maintains genome integrity and therefore suppresses tumorigenesis [52].

Mutations of core *ATG* genes, including *ATG2B*, *ATG5* and *ATG9B*, are also found in cancers, especially those with high microsatellite instability (MSI), which is characterized by the insertion or deletion in the repeated DNA sequence due to deficient DNA mismatch repair [53,54]. Additionally, the frameshift mutation of UVRAG mentioned above is also found in gastric carcinomas with MSI [47]. In addition to the core *ATG* genes, frameshift mutations in *VPS33A*, which functions during the fusion between autophagosomes and lysosomes [55], are also found in colorectal cancer cells with high MSI [56]. However, how these mutations affect autophagy and tumor progression has not been fully examined. In addition, mutation at an *ATG5* splice site has been reported in a prostate cancer cell line, which prevents ATG12 conjugation and leads to the degradation of ATG12 and ATG16L1, thus inhibiting autophagy [57]. Of note, even though *ATG* genes have a more than 25% mutation rate in gastric and colorectal cancers with MSI [54], a study analyzing 11 different types of cancer supports the idea that core autophagy machinery is not highly targeted by single-nucleotide mutations [58]. This finding could reflect the necessity of an intact autophagy pathway for tumor growth. In line with this, the ablation of autophagy results in more benign diseases rather than invasive cancer [59,60,61,62], indicating that autophagy inhibits tumor initiation but is required for further progression, which will be discussed in the next subsection.

Most of the studies mentioned above focus on autophagy activity, but not its selectivity. Recently, by mapping mutations in cancers to LIR-motif-containing proteins, Han et al. found more than 200 potential LIR-motif associated mutations (LAMs) in 148 different proteins [63]. They further examined STBD1, a glycogen autophagy receptor, and found that it inhibits tumor growth and that the cancer-associated LAM inhibits its interaction with LC3 and abolishes its cancer-suppressing capability, drawing a connection between glycogen autophagy and tumor suppression [63]. Other proteins of interest were identified from this analysis. For instance, some core *ATG* genes were identified in the screen, including ATG4B, ATG2B, ATG5 and ATG9A, and the mutation in ATG4B decreases its interaction with LC3. How these mutations affect autophagy and how the potential change in autophagy is associated with cancer need further examination. Additionally, some proteins not in the core autophagy machinery are predicted to contains LAMs, such as BRAF, a defined oncogenic protein [64]. Can these proteins be the cargos or receptors of selective autophagy? Do these mutations in LIR motifs function as a way to escape autophagic degradation and result in cancer? The answer to these questions will shed more light on how autophagy suppresses tumors.

### 2.2. Autophagy and Tumor Promotion

Cancer cells often encounter a stressful environment lacking both nutrients and oxygen, especially in the interior region of a tumor. Because autophagy is an important pathway for cell survival under stress conditions, it is not surprising to see that autophagy supports tumor maintenance and growth. Elevated autophagy is discovered in many types of cancers, suggesting the important role of autophagy in tumor promotion [65,66]. In line with this observation, attenuated tumor growth has been found when autophagy is inhibited [61,62,67,68,69,70,71,72]. One important mechanism by which autophagy promotes tumor growth is to sustain metabolic plasticity [73]. Autophagy can recycle macromolecules and provide metabolic substrates to sustain energy homeostasis [74,75,76] and ablated autophagy in STK11/LKB1-deficient cells leads to insufficient amino acids for mitochondria energy production, excessive fatty acid oxidation and energy crisis [77]. Furthermore, blockage of autophagy also leads to the accumulation of ROS, resulting in DNA damage, which is another possible reason for impaired cancer cell growth after autophagy inhibition [67].

Importantly, even though tumor promotion through autophagy is discovered in many cancers, it depends on the genotype of the cancer cells, or more specifically, the expression of TP53/p53. In both breast cancer and non-small-cell lung cancer, inhibition of autophagy only impairs the growth of the cancer cells expressing TP53 [78,79], indicating that TP53 inhibits tumor growth when autophagy is inhibited. However, whether TP53 determines the consequence of autophagy inhibition is controversial when it comes to pancreatic ductal adenocarcinoma [60,80]. Overall, it seems unlikely that inhibiting autophagy can be used as a general therapeutic approach, but there may be patient-specific applications depending on the genotype of the cancer cells.

### 2.3. Autophagy and Tumor Metastasis

Metastasis is one of the hallmarks of cancer and causes most cancer-related deaths [81]. Increased expression of autophagy associated genes is correlated with a more aggressive and invasive phenotype [82,83], suggesting that autophagy primarily promotes tumor metastasis.

#### 2.3.1. Autophagy and Cancer Cell Motility

The very first step of metastasis is to gain motility and invasiveness to allow migration from the primary tumor site. Cell migration is critical at the early stage of metastasis and turnover of focal adhesions (FAs) is important during this process [84]. FA turnover is promoted by autophagy, though degrading FA proteins such as PXN (paxillin) [85,86]. Additionally, a recent study shows that the deletion of *RB1CC1* or *ATG5* leads to different FA morphologies, but they both result in reduced cell motility [87]. However, during energy starvation, RB1CC1 is activated by ULK1 and inhibits focal adhesion kinase PTK2/FAK, leading to reduced cell motility and tumor metastasis [88]. Together, these studies suggest that energy-starvation-induced autophagy may inhibit cancer cell metastasis, but basal autophagy is necessary for cancer cells to gain motility and invasiveness.

NBR1 is also suggested to mediate the autophagy-dependent disassembly of FA proteins through its interaction with FA proteins and LC3 [89], suggesting the selective degradation of FA proteins. However, which FA protein or proteins are the cargo of NBR1-mediated selective autophagy is unknown. Additionally, because NBR1 contains a ubiquitin-binding domain, ubiquitination of FA proteins may participate in this process, but additional studies are needed to confirm this hypothesis. In a recent study, researchers from Jayanta Debanth’s lab reported that in breast cancer, autophagy inhibition promotes aberrant NBR1 accumulation, which induces the expression of basal epithelial markers and metastatic outgrowth [90]. Together, these studies indicate that autophagy may have distinct roles at different stages of metastasis, supporting cell matrix detachment in an early step, but suppressing metastatic outgrowth at later stages, and NBR1 seems to be a critical protein in both types of regulation. This idea is in line with some discoveries that metastases have a higher level of autophagy than the primary tumor cells, and early cancer metastases have the highest LC3 level [91]. However, many questions remain to be answered: What are the downstream targets of NBR1 when it promotes metastatic outgrowth? Overexpressing NBR1 promotes SQSTM1 accumulation and phosphorylation in liquid-like bodies [92] and SQSTM1-mediated activation of NFE2L2/NRF2 provides hepatocellular carcinoma cells proliferation potency [93]. In addition, aberrant regulation of NFE2L2 is correlated with high-level resistance to anticancer drugs [94]. Therefore, NBR1 accumulation by autophagy inhibition may help tumor metastases outgrowth through SQSTM1 and NFE2L2, but this still needs further investigation. Additionally, how NBR1 is regulated and plays different roles in cancer cell motility and metastases outgrowth and whether NBR1 is involved in other cancers need further investigation.

Epithelial-mesenchymal transition (EMT), a pro-metastatic process wherein epithelial cells gain mobility and invasiveness to become mesenchymal stem cells, is also regulated by autophagy [95]. Autophagy, on the one hand, promotes EMT [96,97,98,99]; however, how this happens remains controversial. Some studies indicate that autophagy regulates EMT in a TGFB-dependent manner [96], but others point out that TGFB-induced EMT partially depends on autophagy [100]. Therefore, which pathway is upstream remains unclear. Additionally, because TGFB can activate autophagy [101], whether there is a positive loop between these two factors to promote EMT is worth examining. On the other hand, autophagy could inhibit EMT via selective degradation of the transcriptional repressor SNAI1/Snail and inhibiting SQSTM1-dependent stabilization of the transcription factor TWIST1 [102,103,104]. These discrepancies in how autophagy regulates EMT, whether in the same direction but via different mechanisms or even in opposite directions, indicate the importance of figuring out if autophagy is directly or indirectly linked to EMT. If directly, does it act in parallel with the known EMT signaling pathways? If indirectly, apart from TGFB, are there interactions between autophagy and EMT pathways that contribute to cancer cell EMT? This is a pertinent questions because, besides the TGFB-SMAD pathway, other types of cell signaling, including PI3K-AKT, MAPK, and RHO GTPase cascades [105] are also key mediators to activate the EMT and they all have close interactions with autophagy [106]; thus, it is possible that autophagy either acts in parallel with one or more of these pathways or they function together to form a network regulating cancer cell EMT.

#### 2.3.2. Autophagy and Resistance to Anoikis

Once cancer cells enter the circulatory system, the disassociation from the extracellular matrix (ECM) leads to anoikis, a form of programmed cell death [107]. When detachment from the ECM occurs, oxidative stress activates EIF2AK3/PERK, which induces autophagy through promoting the expression level of some essential *ATG* genes [108], activating AMPK and inhibiting MTOR complex 1 (MTORC1) [109] and the induced autophagy protects the cell from anoikis [108,110]. Besides EIF2AK3, IKK also induces autophagy but via an MTORC1-independent pathway [111]. In a recent study, MTDH/AEG-1 is reported to promote anoikis resistance, which partially depends on autophagy [112].

Even though different signals have been revealed to induce autophagy after ECM detachment, the mechanism downstream of autophagy induction remains unclear. Proteins from the BCL2 family are key players in anoikis [113] and the close connection between autophagy and apoptosis implies that autophagy may regulate anoikis through BCL2-family proteins [114]. During anoikis, BCL2L11/BIM, a BCL2-family pro-apoptotic protein, is translocated to mitochondria and promotes the assembly of BAX-BAK1 oligomers to induce apoptosis [113]. BCL2L11/BIM can form a complex with BECN1 [115]. Therefore, upon autophagy induction, the increasing level of BECN1 may be able to sequester BCL2L11 from mitochondria, thus inhibiting cell death. Beyond the direct interaction with these BCL2-family proteins, autophagy may indirectly inhibit anoikis. MAPK1 is critical for apoptosis prevention [116] and autophagy may activate MAPK1 function through phosphorylation [117]. Therefore, it is possible that autophagy promotes anoikis resistance through its regulation of MAPK1; however, the precise mechanism needs further examination.

### 2.4. Autophagy and Cancer Stem Cells

Cancer stem cells (CSC) are a subpopulation of cancer cells, which are similar to normal stem cells, but proposed to be critical for tumor metastasis because of their high mobility and self-renewal ability [118]. Autophagy is upregulated in a variety of CSCs and found to be important for CSC survival and maintaining stemness, which encompasses the fundamental properties of stem cells such as self-renewal and generating daughter cells [119,120,121]. Previously, IL17B was shown to be overexpressed in breast cancer tissue and inversely correlate with breast cancer patient survival rate [122], and a recent study indicates that IL17B induces autophagosome formation in gastric cancer CSCs; in contrast, autophagy inhibition through *ATG7* deletion inhibits IL17B-induced self-renewal, which draws a connection between autophagy and CSC maintenance [123]. The ability of autophagy to maintain CSCs may provide an explanation as to why some *ATG* genes, such as *LC3B*, *GABARAP* and *ATG5*, have been found to correlate with poor prognosis [124,125].

Autophagy supports CSC stemness through several downstream pathways. First, in breast cancer, autophagy supports stemness through inducing the secretion of IL6 [126], which is important for stemness maintenance [127]. Second, EGFR-STAT3 and TGFB-SMAD signaling pathways are found to act downstream of autophagy to sustain CSC stemness [128]. Third, in gastric CSCs, a higher expression level of FOXA2 is sustained by autophagy [129], which promotes cell proliferation and maintain CSC stemness [130], and overexpressing FOXA2 partially rescues the decreased self-renewal ability when autophagy is inhibited [129]. Finally, autophagy can augment cell stemness through degrading ubiquitinated TP53 [125,131]. All the studies mentioned above indicate that autophagy positively regulates CSC. However, this observation does not mean that continually higher autophagy activity is better for CSCs. A study from Shashi Gujar’s lab demonstrates that autophagy promotion and suppression both result in a decrease in pluripotency and an increased differentiation or senescence of CSCs [132], indicating that a proper autophagy level is essential to sustain the stemness of CSCs.

### 2.5. Autophagy and Dormant Cancer Cells

One reason tumors are difficult to treat is the existence of dormant cancer cells, which refers to some cancer cells that have arrested growth but are able to retain proliferative capacity and lead to subsequent tumor growth [133,134]. Even though dormant cells have similarities to CSCs such as drug resistance, fundamental differences exist. CSCs are considered as “slow cycling cells” whereas dormant cancer cells undergo cell cycle arrest. In addition, CSCs express stemness marker genes and are at the apex of the differentiation hierarchy. However, dormant and activated cancer cells are at the same differentiation stage and the switch between dormancy and activation is reversible [135].

Many independent studies show that under different dormancy induction conditions, cancer cells show a higher autophagy activity [136], suggesting that autophagy may be critical to maintain the survival of these dormant cells. A majority of ovarian cancer patients develop tumor recurrence, possibly due to the existence of dormant cancer cells [137], and therefore, ovarian cancer becomes a major model to study cancer dormancy. DIRAS3/ARHI usually has a lower expression level in ovarian cancer cells, but the re-expression of DIRAS3 induces autophagy and these cells keep dormant when they grow in a mouse model. The inhibition of autophagy through chloroquine (CQ) treatment leads to a reduced regrowth of these dormant cells in the mouse model, indicating that DIRAS3-induced autophagy is critical for the survival of dormant cancer cells [138,139]. Another study shows that AKT inhibition induces dormancy-like ovarian cancer cells. Under this condition, autophagy activity is increased, and autophagy inhibition will reduce cell viability [140]. Interestingly, DIRAS3-induced autophagy seems to play a different role in vitro and in vivo. As mentioned previously, expressing DIRAS3 in xenograft leads to dormant cancer cells, whereas in cell culture, autophagy induction through DIRAS3 leads to cell death [141]. This difference may come from a more complicated in vivo system than the in vitro cell culture and the interaction between the cancer cell and its microenvironment may contribute to the choice between apoptosis or quiescence. Beyond ovarian cancer, cancer cell dormancy promotion by autophagy is seen in many other cancer types as well. Dormant breast CSCs are highly autophagic [142], which supports cell survival during dormancy [143]. In glioblastoma, autophagy reprograms cancer cell metabolism and promotes cancer cell quiescence [144].

Inhibition of autophagy facilitates cancer cell escape from the dormant state [142,145]. PFKFB3, which may promote cancer cell metastasis [146], could be a key factor in this process. PFKFB3 is an autophagy substrate and an elevated level of PFKFB3 when autophagy is impaired may explain the induced tumor recurrence [142]. Recently, *MIR27A* was reported to ameliorate chemoresistance of breast cancer cells and, at the same time, inhibit autophagy [147], further suggesting that impairment of autophagy leads to cancer cell emergence from the dormant state.

Whether autophagy could be a target to deal with dormant cancer cells having high drug resistance needs a more careful examination. Is awakening dormant tumors via inhibiting autophagy a good way to make them more sensitive to chemotherapy and preventing tumor recurrence? A recent study suggests that treating DIRAS3-overexpressing ovarian dormant cancer cells with crizotinib further increases autophagy and induces apoptosis, and treating mice carrying DIRAS3-expressing ovarian cancer cells with crizotinib prolongs life span [148], even though, as mentioned above, DIRAS3-indcued autophagy supports dormant cancer cell survival. These findings suggest that autophagy may have dual roles in sustaining tumor dormancy and that inducing autophagy to an improperly high level may also become a way to eliminate dormant cancer cells.

### 2.6. Autophagy and Cancer Therapy

As noted above, autophagy can suppress or promote tumors, indicating that modulating autophagy could be a way to treat cancer. CQ and its derivate hydroxychloroquine (HCQ), which impairs autophagosome fusion with a lysosome [149], are two autophagy inhibitors that have been approved by the FDA. CQ treatment inhibits tumor growth both in vitro and in vivo [150,151,152]. Additionally, CQ/HCQ sensitizes cancer cells to chemotherapy, so they are usually used together with other drugs [153], such as temozolomide in solid tumors and melanoma [154], bortezomib in myeloma [155], and gemcitabine in pancreatic ductal adenocarcinoma [156]. Of note, autophagy-independent effects may occur when treating cancer cells with CQ, but these effects still suggest CQ may be a promising compound to treat cancer. For instance, CQ-induced blood vessel normalization, which does not rely on autophagy, restrains tumor invasion and metastasis [157].

Even though the clinical trial of CQ/HCQ is promising, there are still many concerns or limitations in CQ/HCQ application. First, a study culturing three different human cancer cell lines at acidic pH found that autophagic flux is not inhibited by CQ under this condition [158], suggesting that CQ may be not effective in the tumor region with an acidic microenvironment. Second, the results of CQ administration may be different depending on the tumor background. For instance, Maycotte et al. found that breast cancer cells with high STAT3 activity are more sensitive to autophagy inhibition than those with lower STAT3 activity [159], indicating CQ application may not be effective in some breast cancer subtypes. Third, as mentioned, the role of autophagy in tumor growths depends on cancer cell genotype [80], which may potentially lead to different CQ/HCQ effects.

Several inhibitors targeting the early steps of autophagy are also proposed to be potential drugs to treat cancers (Figure 2), such as the ULK1 inhibitor SBI-0206965 [160], the PtdIns3K inhibitors SAR405 and SB02424 [161,162] and the ATG4B inhibitors UAMC-2526 [163], NSC185058 [164], and S130 [165]. These molecules have not gone into clinical trial, but their induction of cancer cell death or sensitivity to chemotherapy highlights their potential to treat cancers.

Because of the dual roles of autophagy in cancer, some autophagy activators are also suggested in cancer therapy. Rapamycin, a commonly used MTOR inhibitor has been proposed to inhibit tumor growth in many cancer types [166,167]. Palbociclib, an AMPK activator, is reported to increase both autophagy and apoptosis in hepatocellular carcinoma [168]. However, the usage of autophagy activators is not as common as that of inhibitors [153].

Even though the studies summarized above show attractive and promising results for the use of autophagy inhibitors or activators to treat cancer, important questions remain. First, it is not known which step of autophagy should be modulated to achieve a better result. Similarly, which autophagy-associated protein is the best target should be carefully examined because these proteins are also involved in other processes. For example, PIK3C3, PIK3R4 and BECN1 can form complexes with either ATG14 (complex I) or UVRAG (complex II), so PIK3C3 is involved not only in autophagy but also endosomal trafficking [169], and the PtdIns3K inhibitor SAR405 compromises both pathways [161]. To alleviate the risk of side effects due to the simultaneous influence on other pathways, drugs targeting more autophagy-specific proteins are needed. Recently, through a screening of more than 2000 molecules, researchers found 19 molecules that interrupt the interaction of BECN1 and ATG14, inhibit the formation of complex I and impair autophagy but not endosomal trafficking [170]. Whether these chemicals, which are more specifically targeting autophagy compared with the other PIK3C3 inhibitors, could have better effects on cancer therapy is worth further assessing. Autophagy also has a complicated crosstalk with other pathways, including cell death and cell migration. Therefore, the modulation of autophagy may lead to alternations of these fundamental cellular pathways, leading to a lower efficiency in tumor elimination or side effects.

Second, modulating autophagy may have different impact on cancer cells and other cells in the tumor microenvironment. While inhibiting autophagy may slow cancer cell growth, autophagy is also needed for other cells in the tumor microenvironment, especially for immune cell development and function [171,172]. Therefore, an expected result may not be achieved if these autophagy inhibitors are simply targeted to the general area of a tumor due to their influence on other neighboring cells.

Third, autophagy plays different roles during the progression of tumors, and even within metastasis, as mentioned above, autophagy has dual roles, inducing detachment but inhibiting outgrowth [89,90], suggesting that it may not be a good idea to use only one compound that either induces or inhibits autophagy. Therefore, different drugs may be used based on the tumor stage and the background of the cancer cells.

Finally, whether cancer cells can develop autophagy inhibition resistance is worth considering. In the clinical trial of the combination of CQ and bortezomib, 45% of the patients are reported to have a period of stable disease [155], suggesting that cancer cells may develop some compensatory mechanism to deal with autophagy inhibition. Two recent studies indicate that knocking out several *ATG* genes, in both cancer cells and normal cells, leads to an increased expression of NFE2L2, a master transcription regulator, which increases proteasome production even though those proteosomes are not as effective as those in WT cells possibly due to the lack of autophagy [173,174]. The upregulation of the proteasome pathway may possibly make up for the loss of autophagy to maintain homeostasis. These autophagy-deficient cells with high NFE2L2 expression are sensitive to proteasome inhibition [173], but the clinical trial mentioned above involves the combination of autophagy and proteasome inhibitor [155]. Therefore, it is possible that some other pathways are also altered to circumvent the loss of autophagy. There are not many studies focusing on whether or how cancer cells compensate for autophagy inhibition, but exploring further into this aspect will help us optimize the efficiency and evaluate the long-term effect of autophagy-based cancer treatment.

## 3. Autophagy and Neurodegenerative Diseases

One hallmark of neurodegenerative diseases is the abnormal accumulation of certain neuroproteins. Because autophagy is critical for the degradation of protein aggregates and maintaining cellular homeostasis, it is not surprising to see that autophagy has a close connection with neurodegeneration: autophagy is responsible for the clearance of accumulated proteins, and this role is particularly important in non-dividing cells. In this section, we will discuss the role of autophagy in Parkinson, Alzheimer, and Huntington diseases.

### 3.1. Parkinson Disease

PD is characterized by the progressive loss of dopaminergic neurons of the substantia nigra, which is accompanied by the accumulation of SNCA/α-synuclein in the form of Lewy bodies and Lewy neurites [175]. From genome-wide association studies (GWAS), great advances have been made in recent decades with the identification of monogenetic causes of PD, including mutations in *SNCA*, *LRRK2*, *PRKN*, and *PINK1* [176,177].

SNCA is a substrate of CMA [178], and, consistent with this fact, boosting CMA decreases SNCA levels and protects cells from wild-type SNCA-induced neurotoxicity [179]. The SNCA accumulation and neurotoxicity in PD patients may result from two factors related to CMA. First, PD-associated mutant SNCA, A53T and A30P, are degraded by CMA less efficiently because they bind to the lysosome but cannot be translocated into the lysosomal lumen, which, at the same time, inhibits the degradation of other CMA cargos and increases cell toxicity [180,181]. Second, the expression level of essential CMA proteins, such as LAMP2A and HSPA8, decreases significantly in PD patient brains [182,183,184]. In addition, the PD-associated UCHL1^I93M^ mutation facilitates interaction with LAMP2A, thus inhibiting CMA [185].

Besides CMA, SNCA is degraded through macroautophagy in neuronal cells [178] and particularly, via selective-autophagy mediated by SQSTM1 as the receptor in microglia [186]. At the same time, SNCA regulates autophagy. Overexpression of SNCA inhibits autophagy via RAB1, which further leads to the mislocalization of ATG9 [187]. Overexpression of PD-associated mutant SNCA^E46K^ impairs autophagosome formation through the inactivation of the MAPK8/JNK1-BCL2 pathway [188]. One recent study, expressing human SNCA in Drosophila, found that SNCA impairs macroautophagy through stabilizing the actin cytoskeleton, which inhibits the fusion between lysosomes and autophagosomes [189]. Even though SNCA is a target of autophagy, SNCA aggregates are not easily degraded by autophagy and inhibit this process by impairing autophagosome clearance [190]. Autophagy is not only responsible for the degradation of SNCA, but also affects its cell-to-cell transmission [191]. Several studies indicate that the blockage of autophagy induces SNCA secretion through exosomes [192,193,194], which reduces cell death, but creates a microenvironment with an inflammatory and neurotoxic response [194]. Additionally, the secreted SNCA will be taken up by other neurons and act as a seed for aggregation in the recipient cells [195].

LRRK2 mutations are one of the most common causes of PD. In most cases of LRRK2-associated PD, the protein has the G2019S mutation and the cells display the SNCA aggregates as Lewy bodies and undergo cell death [196]. Similar to SNCA, LRRK2 is another substrate of CMA, but the LRRK2^G2019S^ mutant is again difficult to degrade and inhibits CMA, which underlies the toxicity in PD by compromising the CMA-mediated degradation of SNCA [197,198]. Besides, LRRK2^R1441G^, which leads to age-dependent SNCA accumulation, also inhibits CMA [198]. LRRK2 regulates macroautophagy as well, but the role remains undetermined. Many studies, involving LRRK2 kinase inhibitor and LRRK2^G2019S^, which has higher kinase activity, demonstrate that LRRK2 inhibits autophagy [199,200]. In contrast, some studies indicate that LRRK2 may promote autophagy through the activation of the MAP2K/MEK-MAPK/JNK-MAPK/ERK pathway [201] and it is reported that age-dependent dopaminergic neurodegeneration and autophagy impairment occur in *lrrk1 lrrk2* double-knockout mice [202]. Further studies should integrate these relevant findings and draw a more complete model of how LRRK2 affects autophagy, which will be of great significance in designing autophagy-targeting PD therapy.

Mitochondria dysfunction has long been recognized as the initiating factor in dopaminergic neuronal loss [203]. Of note, mutations in PINK1 and PRKN, two critical proteins in mitophagy, are highly associated with PD [204]. Interestingly, PD-associated PINK1 mutations are clustered in the kinase domain [204] and several mutations such as G309D, L347P and W437X have a compromised interaction with PRKN, thus inhibiting mitophagy execution [205]. In addition to the mutations in these two proteins, studies focused on other PD-associated proteins also shed light on the importance of mitophagy in PD. Pathogenic SNCA impairs mitochondrial function via binding to OMM proteins such as TOMM20, which impairs protein import to mitochondria [206,207,208], or decreasing the mitochondrial SIRT3 level [209]. As mentioned above, mitophagy is responsible for impaired mitochondria degradation and a study expressing SNCA in yeast shows that Sir2-mediated mitophagy is induced and the selective degradation of mitochondria is responsible for the SNCA toxicity [210]. However, in neurons or in in vivo models, how SNCA-mediated mitochondrial damage is related to or affects mitophagy is unknown. Compared with SNCA, there are more studies about LRRK2 and mitophagy. It is found that the PD-associated LRRK2^G2019S^ mutation inhibits mitophagy by affecting mitochondria motility [211], inhibiting mitochondrial fission [212], and phosphorylating RAB10 to inhibit its mitochondrial accumulation and interaction with OPTN [213].

Here, we focus on SNCA, LRRK2, PINK1 and PRKN, summarizing how they interact with autophagy. Other proteins with PD-associated mutations, such as VPS35, VPS13C and FBXO7, have been suggested to play a role in autophagy (Table 1).

GWAS provides us with invaluable information to study the connection between PD and autophagy and identify therapeutic targets. However, with 90 variants nominated as PD-related factors [176], how to study them, particularly how they are related to autophagy, needs further consideration. First, some PD-associated genes are only studied by knockout instead of using the PD-associated mutated form (such as *VPS13C* [216]). Even though studies of mutant proteins may lead to the concern that the point mutation may not be sufficient to result in either an autophagy or pathological phenotype, further studies focusing on the mutation may shed light on a more detailed mechanism of PD and autophagy. Second, some PD-associated genes discovered through GWAS studies may affect autophagy, but few studies delve into them with regard to mechanism. For instance, several genes, such as *CHCHD2* [228] and *ATP13A2* [229,230], are critical for mitochondrial quality, but whether they have any connection with mitophagy remains unclear. Third, controversial data exist, which may have resulted from the use of different cells lines, and the phenotype at the cellular level is sometimes different from that at the behavioral level [231]. Therefore, based on the goal of specific studies, the model used to study these genes and what marker(s)/phenotype(s) should be used as an indication of PD need consideration as well.

### 3.2. Alzheimer Disease

Alzheimer disease (AD) is a progressive neurodegenerative disease characterized by cognitive impairment and loss of memory. AD patients usually show the accumulation of misfolded proteins such as amyloid-β (Aβ) and hyperphosphorylated MAPT (microtubule associated protein tau) [232].

Several lines of evidence indicate deficient autophagy in AD patients, which include the decreased level of autophagy-related genes, including *BECN1* [233], *ATG5* and *LC3B* [234], and the accumulation of autophagosomes [235]. More importantly, the accumulation of autophagosomes correlates with AD pathology [236], which further indicates the importance of understanding the connections between autophagy and AD. Besides these direct lines of evidence, AD-associated mutation in PSEN1 disrupt autophagy [237,238]. A decreasing level of PICALM, which occurs in AD, inhibits autophagy and exacerbates AD pathology [239,240,241]. However, some studies report an increase in autophagy when cells are treated with Aβ [242]. This discrepancy could at least in part be a consequence of the AD stage. In 2016, Bordi et al. carried out a comprehensive analysis at different stages of AD, finding an upregulation of autophagy-related genes at the early stage, but an impeded autophagy flux at the late stage [243]. The mechanism of this change is not clear, possibly because autophagy is induced at the early stage to degrade the protein aggregates. However, at the later stage, autophagy or lysosome clearance ability becomes inhibited by the accumulation of abnormal proteins.

The relationship between Aβ and autophagy is complicated. First, Aβ is degraded through autophagy, and several studies show a decreased Aβ level in cells and improved cognitive ability in an AD mouse model when autophagy is induced [244,245,246,247]. Second, Aβ may also be generated inside autophagosomes because both APP (amyloid beta precursor protein) and PSEN1, an enzyme involved in the cleavage of APP to form Aβ, are found within the autophagosome [235]. Third, one study reported that the secretion of Aβ to the extracellular space, where plaque forms, depends on autophagy in neurons [248]. On the contrary, a recent study indicates that MTORC1 inhibition reduces amyloid secretion due to the upregulation of autophagy [249]. Interestingly, the activation of AMPK does not induce autophagy in neurons, and different AMPK activators results in differential regulation of Aβ secretion, either increasing or reducing, which indicates a complex role of AMPK in Aβ secretion independent from autophagy [249]. 

APP has a KFERQ motif, which is typically associated with CMA. However, the deletion of this motif in APP does not abolish its interaction with HSPA8, but conversely, increases the interaction [250]. The authors raise the possibility that the KFERQ motif may be used to bind to AP2, which is an autophagy adaptor, because the AP2 recognition sequence is part of KFERQ; deletion of the KFERQ motif impairs AP2-dependent targeting to the lysosome. Further studies should investigate the binding partner of the KFERQ motif other than HSPA8; this may lead to the identification of new functions of this motif other than acting as the CMA signal and shed more light on AD pathology. 

The other hallmark protein in AD, MAPT, is a substrate of macroautophagy, CMA and microautophagy, but some AD-associated MAPT mutations cannot be cleared efficiently by autophagy [251,252]. Consistently, activation of autophagy through inhibiting MTORC1 helps with prolonged clearance of MAPT [253]. CMA is downregulated in AD patient brains [254]; CMA upregulation improves the disease phenotype that results from MAPT or combined MAPT and Aβ pathologies, and inhibition of CMA accelerates the AD pathology in a mouse model [254].

Together with Aβ and MAPT, compromised mitochondria accumulation is another hallmark of AD. Although it is unclear whether the mitochondrial dysfunction is a cause or a consequence of Aβ and phosphorylated MAPT accumulation [255], it indicates that quality control of mitochondria is impaired in AD neurons. Deficient mitophagy has been discovered in AD patient brain and patient stem-cell derived neurons [256]. The compromised mitophagy in AD may resulted from the following. First, in AD patient brains, PINK1-PRKN-dependent mitophagy is enhanced with Aβ accumulation, but it is followed by a progressively depleted PRKN [257,258], suggesting that mitophagy is induced at the early stage of AD, but finally shows an inadequate capacity compared with the huge number of damaged mitochondria. Additionally, higher levels of Δ1 PINK1, the main cleaved product of PINK1, is found in AD patient brain, which inhibits PRKN translocation to mitochondria and impairs mitophagy [258]. Second, emerging data are showing that Aβ and phosphorylated MAPT interfere in the mitophagy pathway and MAPT impairs PRKN translocation to mitochondria [259,260,261]. Overall, these studies demonstrate that compromised mitophagy and abnormal mitochondrial dynamics contribute to AD pathogenesis.

### 3.3. Huntington Disease

Huntington disease (HD) is characterized by cognitive dysfunction, uncontrolled movement, and alternation in mood [262]. HD is caused by the accumulation of a mutant (m) form of HTT (huntingtin), which has abnormally long tracts of polyglutamine (polyQ) repeats [262].

mHTT aggregates can be degraded through selective autophagy, with the help of SQSTM1 [263] and OPTN [264] as the receptors, indicating autophagy is critical to prevent HD. Interestingly, a study shows that the depletion of SQSTM1 leads to a longer life span in a mouse model with mHTT [265]. The data suggest that the aggerated mHTT is localized in both the nucleus and cytoplasm. *SQSTM1* deletion only affects the protein degradation in the cytosol but not the nucleus, and the reduced nuclear inclusion may result from the inhibited nuclear transport of mHTT from the cytosol [265]. However, in this study, the inhibition of autophagy through *ATG5* knockout is toxic, still indicating the importance of autophagy in protein aggregate clearance and inhibiting HD pathology [265].

Autophagy activity is altered in HD. It has been reported that autophagy-associated gene expression is changed in HD patients [266]. Even though some genes at the early autophagy stages are upregulated [266], autophagy activity is impaired because of the deficient capability of trapping cytosolic cargos [267]. Additionally, mHTT leads to decreased phosphorylation of ATG14 by ULK1, which further leads to the reduction in PtdIns3K kinase activity and autophagy [268]. Therefore, not surprisingly, inducing autophagy either through the MTORC1-dependent pathway or MTORC1-independent pathway can reduce the toxicity of mHTT and ameliorate the HD phenotype [269,270,271]. 

Of note, conformational polymorphism of polyQ proteins has been suggested [272] and the polyQ antibody 3B5H10-recognized mHTT is the most toxic form; it has a slower degradation rate because of the resistance to K63 ubiquitination and SQSTM1 recognition [273]. It also raises some questions to consider. For example, is there a conversion between different conformations? Can we convert the ones that are the most resistant to degradation to more autophagy sensitive forms? The answers to these questions may lead us to think about new ways to ameliorate mHTT accumulation to treat HD.

Even though mutant HTT leads to protein aggregates and neurotoxicity, wild-type HTT, in contrast, helps autophagy. The role of HTT as a scaffold protein in selective autophagy is suggested by a similar structure between the C-terminal domain of HTT and yeast Atg11 and the capability of binding with key Atg11 interactors [274]. Further study confirms the interaction between HTT and SQSTM1 to facilitate its connection between LC3 and ubiquitinated cargo [275]. Additionally, this study also shows that HTT binds to ULK1, releasing it from the negative regulation by MTORC1, to reach the maximal activation of selective autophagy, and this modulation only occurs when selective autophagy is specifically induced but not during starvation-induced autophagy [275]. This elegant study reveals a positive role of HTT in selective autophagy, but also raises several questions. How is HTT regulated to distinguish between selective autophagy and starvation-induced autophagy? To what extent does the HD-associated mutation affect normal HTT function? The answers to these questions will bring us a better understanding about the relationship between HTT and autophagy and may shed light on new autophagy-based therapeutic approaches to HD.

### 3.4. Autophagy and Therapy of Neurodegenerative Diseases

As mentioned above, autophagy can degrade toxic protein aggregates, which are the major causes of neurodegenerative diseases, raising the possibility that modulation of autophagy could be a possible therapeutic approach to the treatment of such diseases. Potential autophagy-targeting therapeutic approaches can be divided into three categories. First, generally stimulating autophagy may help in the clearance of aberrant protein aggregates and improve the symptoms of the pathology. Multiple autophagy-inducing small molecules have been tested in the mouse model [276]. For instance, rapamycin and its analog temsirolimus lower the level of MAPT [277] and rescue memory impairment in an AD mouse model [278]. In addition, rapamycin is also able to improve the motor ability of both PD and HD mice [269,279] and to revert the cognitive and affective deficits of PD mice [280]. Trehalose, an AMPK activator, not only induces autophagy and improves the clearance of SNCA and motor function in PD mice injected with SNCA^A53T^ [281] but improves HD pathology as well [282]. Furthermore, applying the combination of rapamycin and trehalose to the PD mouse model shows an additive effect on dopaminergic deficits [283]. In addition, felodipine, a calcium channel blocker and autophagy inducer, can stimulate the degradation of SNCA in PD mice and improve the motor ability of HD mice [284]. S14G-humanin and melatonin can alleviate the accumulation of Aβ in the brain [285] and prevent cognitive decline [286] through inducing autophagy in AD mice. Besides the chemicals tested in the mouse model, lithium and metformin are being used in clinical trial in patients with mild cognitive impairment, which is associated with high risk of PD, and the studies indicate that these two autophagy-stimulating molecules can attenuate cognitive decline in these patients [287,288]. Nilotinib, an AMPK activator, has also been used in a clinical trial of PD patients [289]. 

Second, mitophagy dysregulation is also highly associated with neurodegenerative diseases; therefore, molecules modulating mitophagy may improve neurodegenerative pathologies. This idea has been tested in an AD mouse model. Treating the AD mice with urolithin A, a mitophagy inducer, inhibits MAPT hyperphosphorylation and improves memory [256]. Recently, through a screening of more than 2000 FDA-approved drugs or drug candidates, Cen et al. found UMI-77 as a mitophagy inducer and treating the APP/PS1 mouse model with UMI-77 reduces the brain insoluble Aβ level [290]. 

Third, specifically attaching neurodegeneration-associated proteins to LC3 or HSPA8 can induce their degradation through autophagy or CMA, respectively. In 2010, a 46 amino-acid peptide adaptor molecule was designed, which contains an expanded polyQ tract-binding motif and two HSPA8-binding motifs. This molecule is sufficient to reduce polyQ aggregation, ameliorate symptoms and extend the life span of HD mice [291]. Recently, adapting the idea of autophagic degradation of mHTT, researchers found small molecules that interact with mHTT and LC3 at the same time, which are sufficient to bring mHTT to the phagophore, reduce the mHTT level and alleviate the HD-related phenotype [292]. Similarly, a cell membrane-penetrating peptide, which specifically binds the Aβ oligomer and contains three CMA-targeting motifs can reduce the Aβ oligomer level and protect neurons from Aβ oligomer-induced neurotoxicity [293]. However, this peptide has only been tested in neurons, and not in an AD mouse model. Therefore, further experiments are needed to demonstrate its practicality in vivo. 

Even though the studies mentioned above indicate intriguing results of autophagy-targeting therapy, the complicated connection between autophagy and neurodegenerative diseases requires us to get an even deeper understanding before we could apply these approaches efficiently. The complexity of this relationship first come from the various disease-associated mutations that may affect different steps of autophagy. If the mutant protein inhibits the later steps of autophagy, such as the degradation step, then inducing autophagy from the early steps may not improve the degradation of protein aggregates and on the contrary may bring side effects because of the increasing accumulation of autophagosomes. Another reason of the complexity is that autophagy not only can degrade proteins aggregates, but also affect their secretion, especially for Aβ in AD as mentioned above. Even though the role of autophagy in Aβ secretion is undetermined, it is worth further study because if autophagy is required for Aβ secretion, promoting autophagy as a treatment of AD may be at the expense of higher extracellular Aβ and plaque formation. Of note, inducing autophagy after plaque formation has no effect on AD-like pathology and cognitive deficits [294], which also indicates that the disease stage to apply autophagy-targeting therapy is critical. Additionally, because autophagosomes are a site where Aβ is generated [235], whether stimulating autophagy will induce the generation of Aβ and if so, whether the upregulated clearance of Aβ can offset the increase in production should be assessed. Because of the complexity, personalized analysis about the cause of the disease and the disease stage may be needed. 

How to evaluate the efficiency of an autophagy-targeting approach also needs serious consideration. One recent study pointed out that impaired autophagy in dopamine neurons in a mouse PD model expressing human SNCA improves motor performance even though the pathology of PD such as progressive neuron loss still exists [231], indicating that while upregulating autophagy is beneficial for cellular pathology, it may lead to an opposite consequence on organismal behavior. Therefore, clinicians and researchers also need to carefully consider whether it is better to examine the cellular or the behavioral level for signs of improved pathology. 

## 4. Autophagy and Infectious Diseases

### 4.1. Autophagy and Bacterial Infection

As mentioned previously, autophagy can be selective, and the invading pathogens can be substrates of autophagy. This selective autophagy is named xenophagy [295]. In this context, autophagy functions as an innate immune mechanism to fight against infectious bacteria. Not surprisingly, bacteria have developed strategies to escape from the autophagic degradation. Here, we will briefly discuss the interaction between autophagy and infectious bacteria (Figure 3).

*Salmonella enterica* serovar Typhimurium (*S. Typhimurium*) is one of the most common model bacteria used to study xenophagy. *Salmonella* invades cells and resides within the cytosol in a modified phagosome termed a Salmonella-containing vacuole (SCV). When the SCV membrane is damaged and the bacterium is exposed to the cytosol, it will be ubiquitinated; this results in the recruitment of several ubiquitin-binding autophagic receptors, such as SQSTM1, OPTN and CALCOCO2/NDP52, which will target the bacterium to the phagophore. The bacterium will finally get trapped in the autophagosome, which fuses with a lysosome for degradation [296]. Additionally, the damaged SCV can be sensed by the vacuolar-type H^+^-ATPase, which then recruits ATG16L1 for autophagosome formation [297]. Damage to the SCV also exposes β-galactoside, which is usually localized on the lumenal surface of endosomes, to the cytosol. β-galactoside can be recognized by LGALS8 (galectin 8), which will then recruit autophagy receptor CALCOCO2 [298]. Similarly, *Mycobacterium tuberculosis* (*M. tuberculosis*) enters the cell through phagosomes. Permeabilization of the phagosome allows the ubiquitination of the bacterium and the subsequent recognition by autophagy receptors such as SQSTM1 and CALCOCO2 [299]. In addition, ubiquitin binds directly to the ubiquitin-associated domain on the bacterial surface protein Rv1468c independently from the E3 ligase and recruits autophagy receptors [300]. Autophagy also selectively targets cytosolic bacteria, such as *Streptococcus pneumoniae* (group A streptococcus). After escaping from endosomes to the cytosol, most of the group A streptococcus will be sequestered within LC3-decorated autophagosome-like vesicles and degraded through fusion with a lysosome [301]. Several RAB GTPases are involved in this process, including RAB7, RAB9A and RAB23, which are responsible for the formation of the vesicles and fusion with lysosomes [302,303]. Interestingly, RAB23 and RAB9A do not participate in starvation-induced autophagosome formation [303]. Additionally, a recent study indicates that the outer-membrane vesicles from Gram-negative bacteria induce xenophagy even before entering the cells through activating AMPK, but non-selective autophagy is not induced [304]. Together, these studies demonstrate a special regulating mechanism of xenophagy but not non-selective autophagy.

Because of the potential degradation through autophagy, many bacteria have evolved mechanisms to hijack this process to allow their survival and replication within infected cells. First, bacteria can inhibit autophagy initiation signaling. For instance, *S.* Typhimurium infection induces lysosomal degradation of AMPK and reactivation of MTORC1, thus inhibiting autophagy [305]. Second, some bacterial proteins can directly interact with autophagy-associated proteins and inhibit their functions. RavZ, a *Legionella pneumophila* effector protein, hydrolyzes the bond between the glycine and the aromatic residue of LC3, thus producing non-functional LC3 that cannot be reconjugated with PE [306]. VirA, a *Shigella flexneri* effector protein, interacts with RAB1 and inhibits its function, thus prohibiting phagophore biogenesis around the bacteria [307,308]. RAB1 is also bound by SseF and SseG secreted by *S.* Typhimurium, which inhibits ULK1 activation and PtdIns3P biogenesis, thus impairing autophagy and supporting the survival and replication of the bacteria [309]. Additionally, in *S.* Typhimurium SopF, a type III secretion system effector, inhibits the recruitment of ATG16L1 to the SCV by the vacuolar-type H^+^-ATPase upon vacuole damage, thus impairing autophagy initiation [297]. Third, several types of bacteria block autophagosome and lysosome fusion, including *M. tuberculosiss* [310], and *Helicobacter pylori* [311]. Besides the three mechanisms mentioned above, bacteria can escape autophagy through masking their surface and through some unknown mechanisms [312]. To summarize, xenophagy is an efficient way to clear bacteria that have invaded into cells and the research focusing on how bacteria escape from autophagy will guide us to design autophagy-targeting approaches to treat infectious diseases.

### 4.2. Autophagy and Viral Infection

Viruses are also the target of selective autophagy [313] and the specific degradation of viruses through autophagy is named virophagy. Several viruses have been demonstrated as the cargo of virophagy, but they can be targeted to this pathway via different mechanisms. For example, LGALS8 detects β-galactosides when picornaviruses are released from endosomes, marking them for autophagic degradation [314]. In addition, hepatitis C virus NS5A (nonstructural 5A) protein, which is critical for HCV RNA replication, interacts with SHISA5/SCOTIN and gets targeted to phagophores, leading to degradation via autophagy [315].

Autophagy also supports virus survival in host cells [316]. Autophagy is needed by some common viruses for their infection and normal replication, including polioviruses [317], hepatitis C virus [318,319], and influenza A virus [320]. The studies summarized above indicate the dual role of autophagy in viral infection, suggesting that targeting autophagy could be a promising way to deal with viral infection.

### 4.3. Autophagy and COVID-19

Among a wide range of human viruses, the one that has recently received the most attention is the severe acute respiratory syndrome coronavirus 2 (SARS-CoV-2), a beta-coronavirus resulting in the ongoing coronavirus disease 2019 (COVID-19) pandemic, which is having a dramatic impact on healthcare and society worldwide. Because the interplay between other coronaviruses from the same family, including SARS-CoV and Middle East respiratory syndrome coronavirus, and autophagy is shown by many previous studies [321,322,323,324], the involvement of autophagy in SARS-CoV-2 infection has been suggested and drugs targeting autophagy have been proposed for COVID-19 treatment [325,326]. This year, many studies have focused on this field, revealing a close but complex connection between SARS-CoV-2 infection and autophagy.

Several large-scale studies, including the use of transcriptomics and proteomics analysis, suggest that SARS-CoV-2 proteins perturb the host cell autophagic pathway [327,328]. More specifically, a systematic functional analysis of SARS-CoV-2 proteins found that ORF3a and ORF7a disturb autophagy flux via different mechanism: ORF3a inhibits the fusion between autophagosomes and lysosomes, whereas ORF7a inhibits lysosome acidification [329]. How ORF3a inhibits fusion is illustrated by two independent studies, which indicate that SARS-CoV-2 ORF3a interacts with VPS39, inhibiting the interaction between the homotypic fusion and protein sorting/HOPS complex and RAB7 or STX17, consequently preventing the formation of a STX17-SNAP29-VAMP8 SNARE complex, and therefore impairing autophagosome-lysosome fusion [330,331]. Another study indicates that ORF3a interacts with UVRAG, thus positively regulating PtdIns3K complex I but inhibiting the formation of complex II. As a result, autophagosome formation can be induced, but maturation is compromised [332]. Why does SARS-CoV-2 choose to inhibit the late steps of autophagy? One possibility is that SARS-CoV-2 uses phagophores or autophagosomes for their replication. Several genetic screens aiming to identify host factors for SARS-CoV-2 suggest the importance of autophagosomes in viral replication [333,334,335,336]. In *BECN1* or *PIK3R4* knockout cells, the replication of SARS-CoV-2 is almost blocked [333]. Additionally, inhibiting the PtdIns3K complex results in a striking decrease in viral load, with some inhibitors reaching a more than 100-fold reduction [334,337]. Interestingly, TMEM41B, an ER-localized protein, is critical not only for the infection of SARS-CoV-2, but other seasonal coronaviruses as well [335,336,338]. TMEM41B has been indicated to participate in autophagy: deletion of *TMEM41B* leads to a stall during autophagosome biogenesis [339,340,341], suggesting that the reason *TMEM41B* deletion inhibits SARS-CoV-2 may be due to the compromised autophagosome formation. Furthermore, double-membrane vesicles are a major coronavirus replication organelle [342,343]; therefore, autophagosomes or phagophores, which are double membrane, are likely to be involved in this process. Additionally, ACE2 (angiotensin converting enzyme 2), the first reported receptor for SARS-CoV-2, contains a LIR motif in its cytoplasmic region, which potentially recruits autophagy elements such as LC3 to facilitate the incoming coronavirus replication [344]. However, on the contrary, some studies report that CQ or HCQ treatment inhibits SARS-CoV-2 infection [337,345,346]. Therefore, whether autophagy facilitates and whether phagophores and/or autophagosomes are sites for SARS-CoV-2 replication need further investigation.

Autophagy may help SARS-CoV-2 escape from the host immune response. One recent study determined that overexpressing one of the SARS-CoV-2 main structural proteins, M, induces mitochondria degradation through mitophagy to block innate immunity signaling and inhibit the type I IFN (interferon) response [347]. Additionally, ORF8 of SARS-CoV-2 mediates the degradation of major histocompability complex class Ι (MHC-I) through autophagy [348], while the specific mechanism and the receptor for this selective autophagic degradation remain unknown.

Another unanswered question is whether SARS-CoV-2 can be degraded through autophagy. It is possible that SARS-CoV-2 inhibits autophagy to protect the virus from degradation. Besides inhibiting autophagy in the fusion step mentioned above, coronaviruses may use the lysosome for non-lytic viral release, which involves lysosomal deacidification and compromised lysosomal hydrolase activity [349]. A recent study points out that SARS-CoV-2 infection results in the accumulation of key metabolites, the activation of autophagy inhibitors and the reduction of several ATG proteins involved in different steps of autophagy including the initiation step [350]. In addition, autophagy induction restricts SARS-CoV-2 growth in primary human lung cells [350], which is consistent with another finding that rapamycin treatment reduces viral replication 4–6 fold [329]. Even though the result of inducing autophagy is not as striking as inhibiting autophagy, as mentioned above, it still suggests the potential role of autophagy in viral degradation.

Despite the studies mentioned above, we must continue to explore the potential dual roles of autophagy in SARS-CoV-2 infection. Having a clearer understanding of the relationship between SARS-CoV-2 and autophagy will help us optimize an autophagy-targeting therapeutic strategy to treat COVID-19.

## 5. Autophagy and Metabolic Disorders

Autophagy, as mentioned above, is a critical process for energy replenishment when the cell is in a stressful environment and is important to maintain the balance of metabolites. In line with this, dysregulation of autophagy has been seen in, and been proposed as one of the causes of, metabolic disorders. Similar to other diseases, the role of autophagy in metabolic disorders is complicated. In this section, we briefly summarize the studies revealing the relationship between autophagy and metabolic disorders, focusing on obesity and diabetes mellitus.

Obesity is one of the most common metabolic disorders around the world and is characterized by the accumulation of dysfunctional adipose tissue when the energy intake exceeds the energy expenditure [351]. Autophagy alternation has been found in many different tissues in obese individuals or in a mouse obesity model. For instance, both genetic or dietary obesity leads to the downregulation of autophagic proteins in liver such as ATG7 [352], and high-fat diet reduces hepatic ATG12, ATG5 and LC3-II levels [353]. In adipose tissue, most studies report an increased autophagy in obese patients [354,355], but high-fructose-induced obesity results in a decreased expression of *ATG* genes [356]. Controversial results are also seen in pancreas and heart [351]. The discrepancy among studies may result from different modes of obesity induction: amino acids, glucose and lipids have different metabolic pathways and may regulate autophagy through alternate pathways [351].

Autophagy is not only changed in an obesity model; dysregulation of autophagy is also suggested as one of the causes of obesity and concomitant metabolic disorders such as insulin resistance (Table 2). Autophagy may be related to obesity from two aspects. First, autophagy is critical in adipogenesis. Multiple *ATG* genes are actively regulated during the adipocyte differentiation [357] and several studies indicate that blockage of autophagy by knocking out essential *ATG* genes impairs adipogenesis [358,359]. Furthermore, some autophagy-regulating proteins or chemicals are reported to modify adipogenesis. For instance, CEBPB/C/EBPβ facilitates adipogenesis via activating ATG4B and promoting autophagy [360]. β-Cypermethrin promotes adipogenesis through inducing autophagy and shaping a microenvironment that is suitable for adipogenesis [361]. However, autophagy is critical and most active only during the initial stage of adipocyte differentiation but not in the later stage [362]. Recently, a study reported decreased RUBCN expression and subsequent upregulated autophagy in aged adipose tissue, which worsens the age-related metabolic disorder [363], indicating that even though autophagy is necessary for adipogenesis, excessive autophagy may not be beneficial for adipose tissue especially during the later stage. Second, autophagy is critical for maintaining the balance between white adipose tissue (WAT), which stores energy, and brown adipose tissue (BAT), which is responsible for thermogenesis [364]. Autophagy facilities brown or beige adipocyte whitening, possibly through the degradation of mitochondria [365,366]. This idea is supported by other studies showing that deficient PRKN helps beige adipocyte maintenance [367], and PRKN-mediated mitophagy is downregulated during the browning of white adipocytes [368]. Additionally, thermogenic activation suppresses autophagy in brown adipocytes, which may help brown adipocyte maintenance and thermogenesis [369]. Therefore, inhibiting autophagy in BAT may become a way to prevent obesity by impairing the conversion from BAT to WAT [370].

Type 2 diabetes mellitus (T2DM) is one of the most common chronic metabolic diseases associated with obesity, which is characterized by the resistance to insulin and inability to produce insulin by pancreatic β-cells [389]. Multiple studies have shown altered β-cell autophagy in mouse obesity or diabetes models and the ablation of autophagy in β-cells leads to decreased insulin secretion and impaired glucose tolerance (Table 2) [390]. T2DM also features the accumulation of IAPP (islet amyloid polypeptide), which is toxic to β-cells through inducing apoptosis. β-cell autophagy inhibition leads to glucose intolerance and loss of β-cell mass in human IAPP-expressing mice [391,392] and inducing autophagy facilitates the clearance of IAPP and improves β-cell activities [393]. All these studies suggest the importance of autophagy in maintaining normal β-cell functions and protecting the individual from diabetes. Autophagy is beneficial to β-cells from different aspects. First, autophagy can protect β-cells from ER stress. Because β-cells have a high demand to produce and secrete proteins, they are quite susceptible to ER stress. ER stress is an inducer of autophagy, which serves a cytoprotective role in β-cells [394,395,396]. Second, ROS is a signal of insulin secretion due to the increasing level of glucose [397], but chronic exposure to ROS leads to abnormal mitochondria, inflammatory damage and finally cell death. Mitophagy in β-cells can serve as an adaptive response to the proinflammatory cytokines and protect β-cells from inflammatory damage [398]. Compromised mitophagy is found in patients with newly diagnosed T2DM and advanced duration T2DM [399]. The impaired mitophagy of diabetes could result not only from the decreased expression of mitophagy genes [399], but the ER-stress and oxidative-stress-induced cytosolic TP53 [400] and also the accumulation of IAPP in β-cells [401]. Moreover, PDX1, which is critical for β-cell mitophagy [402], also has a lower expression level in diabetic models [403]. The fact that mitophagy promotion via overexpressing PRKN or deficient TP53, or sonodynamic therapy can inhibit β-cells dysfunction and maintain insulin secretion in diabetic mice further indicates the importance of mitophagy in preserving the normal function of β-cells [400,404]. Of note, even though autophagy is important for β-cells, too much autophagy may inhibit insulin secretion by degrading insulin granule vesicles [376]. The studies summarized above focus on T2DM, but in type 1 diabetes mellitus (T1DM), β-cell autophagy is also compromised [405]. However, how this impaired autophagy is associated with T1DM needs further study.

Several autophagy-inducing molecules have been tested in a mouse obesity model and the treated mice show ameliorated obesity and associated phenotypes. For example, ADIPOQ/adiponectin stimulates autophagy and reduces insulin resistance caused by high-fat diet (HFD) [406]. Injecting mangiferin, which induces autophagy, into HFD-fed mice improves glucose tolerance and lowers body weight [407]. Liraglutide, a GCG (glucagon)-like peptide-1 (GLP-1) analog, relieves lipid accumulation in the liver of HFD-fed mice through inducing autophagy [408]. More recently, from a screening of small molecules that act as autophagy stimulators, researchers identified MSL, a compound that can reduce the intracellular lipid level; obese mice treated with MSL have a better metabolic profile [409]. In addition to the chemicals that stimulate autophagy, the autophagy inhibitor HCQ protects mice from HFD-induced obesity and insulin resistance [410]. A clinical study also proposed that HCQ is favorable to β-cell function and lowers the risk of T2DM [411].

Even though some drugs targeting autophagy in obesity have been designed, questions remain concerning the practicality of modulating autophagy to treat this condition. First, most of the studies are done in mouse models. However, the living environment and diet of humans is much more complicated, which may result in different consequences compared to mice. Second, autophagy in different tissues responds or contributes differently to metabolic disorders. Can a drug target one specific tissue and modulate autophagy? If not, whether the effect on other tissues can affect the efficiency of autophagy regulation needs to be determined. Third, the specific degradation of lipid droplets through autophagy, lipophagy, is also considered as an important regulator of metabolism. The role of lipophagy is mostly discussed in the context of fatty liver diseases [412], but some studies also show that lipophagy may be related to obesity and diabetes. For instance, a recent study designed autophagy-tethering compounds targeting lipid droplets to induce their degradation. Treating mice with these compounds leads to a lower body weight [413], which indicates a role of lipophagy in obesity. However, the mechanism and the treatment potential of modulating lipophagy need further study.

Besides drug intervention, lifestyle modifications, such as intermittent fasting (IF), are also beneficial for metabolism in the mouse models [414,415] and in humans [416]. The benefits of IF may come from its regulation of autophagy. Fasting can induce autophagy in liver and stimulate lipid degradation [417]. IF can also increase autophagy flux in islets, which enhances β-cell survival and glucose-stimulated insulin secretion [418]. Intermeal fasting is also proposed to induce autophagy and be beneficial to metabolism [419].

## 6. Conclusions and Perspectives

Because autophagy is not only important for energy replenishment during starvation, but also to maintain cellular homeostasis, this pathway has a close connection with human diseases, and the emerging roles of autophagy in disease outcome provide a potential for breakthrough in investigating the mechanism of human diseases and designing therapeutic approaches. In this review, we only summarize the role of autophagy in four types of diseases: cancer, neurodegenerative diseases, infectious diseases, and metabolic disorders, but many other human diseases such as those involving the heart, lungs, kidney, retina, muscle and autoimmunity are also related to autophagy [420] (Figure 4). Even though an increasing number of studies focus on autophagy and human diseases, there are still many questions that need further consideration.

First, using different models and genetic approaches, the modulation of autophagy gives us exciting but sometimes controversial results, indicating the dual roles of autophagy in diseases. The dual role of autophagy could result from studies examining different disease stages and may be inherent to the differences between pathological and physiological autophagy. More investigations are needed to determine what regulates the changing of roles of autophagy, which will provide us with more information when considering modulating autophagy as an approach to overcome diseases.

Second, to what extent we can modulate autophagy needs serious consideration. Even though modulating autophagy seems to be a promising strategy to treat some diseases, autophagy has its own physiological roles and too much or too little autophagy can both be harmful. Deficient autophagy may lead to acute infectious diseases and tissue aging while excessive autophagy could result in cell death. Moreover, autophagy proteins also function in autophagy-independent pathways. The modulation of these proteins may lead to the dysregulation of other pathways, which brings potential side effects. Therefore, autophagy modulators that can be more precisely targeted to disease-related tissues, and targeting more autophagy-specific proteins, should be optimal ways to treat pathologies. 

Third, selective autophagy is closely related to human diseases, such as mitophagy and selective removal of protein aggregates (aggrephagy) in neurodegenerative diseases, xenophagy in infectious diseases, and lipophagy in metabolic disorders. Sometimes, selective autophagy regulation is uncoupled from non-selective autophagy. Therefore, it is possible that we can specifically target selective autophagy rather than non-selective autophagy for disease treatment to reduce the potential side effects of these manipulations. However, how this uncoupled regulation is accomplished and whether there are chemical components/metabolites involved need to be addressed.

Finally, the optimal autophagic markers in the mouse model or human beings are unclear. We have well established guidelines for measuring autophagy in cells [421], but there are few good markers to study autophagy in mouse models and the existing ones have limitations; the situation with humans is even more problematic. These markers are critical for determining how autophagy affects disease initiation or progression and for evaluating the efficiency of autophagy-targeting compounds, especially in a tissue-specific and temporal manner.

Overall, current studies have provided substantial information revealing the emerging roles of autophagy in human diseases. With a deeper understanding of the mechanism of autophagy, we will learn more about how autophagy is associated with human diseases, which will aid in therapy design.

## Figures and Tables

**Figure 1 biomedicines-09-01651-f001:**
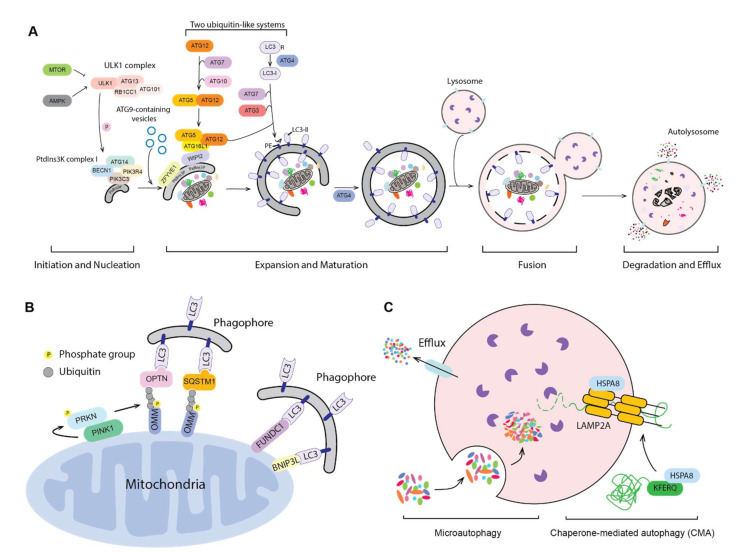
Three types of autophagy. (**A**) The mechanism of macroautophagy. Upon induction, the phagophore nucleates and expands to sequester cytoplasmic substrates randomly or selectively. Upon completion, the phagophore closes and forms a double-membrane autophagosome, which fuses with an endosome (not shown) and/or a lysosome; the autophagic cargos are degraded and the breakdown products are released back into the cytosol. The sequence shown for the LC3B C terminus is for the mouse protein. (**B**) PINK1- and PRKN-mediated and receptor-mediated mitophagy. Upon mitochondrial damage, PINK1 recruits and phosphorylates PRKN on the outer mitochondrial membrane (OMM). PRKN mediated the ubiquitination of OMM proteins, which then bind to autophagy receptors, targeting the mitochondria to a phagophore via interaction with an Atg8-family protein. Additionally, OMM proteins such as FUNDC1 and BNIP3L can serve as the mitophagy receptors. (**C**) Microautophagy and chaperone-mediated autophagy (CMA). During microautophagy, the lysosome membrane rearranges and forms a lumenal vesicle-containing autophagic substrates, which will be degraded within the lysosome. CMA starts from the recognition of a KFERQ-like motif in the substrate protein by HSPA8. HSPA8 facilitates the transport of substrates into the lysosome through the action of LAMP2A and lumenal HSPA8.

**Figure 2 biomedicines-09-01651-f002:**
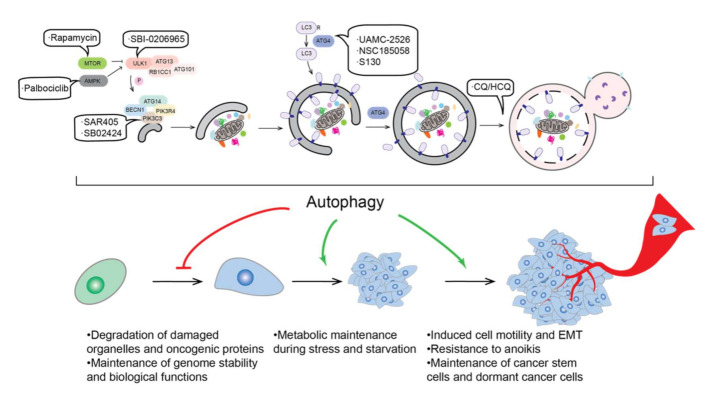
Autophagy in tumor progression and autophagy-targeting cancer therapy. Autophagy has been proposed to inhibit malignant transformation through different mechanisms, including the degradation of damaged organelles and oncogenic proteins and the maintenance of genomic stability and biological functions. After the tumor forms, autophagy basically supports tumor progression mainly through regulating metabolic process during stress or starvation. Tumor-supporting functions of autophagy are also reflected from its ability of promote cell motility, EMT and resistance to anoikis and to maintain cancer stem cells and dormant cancer cells. Because controversial conclusions exist, we only present what most studies indicate. Autophagy modulation could have the ability to treat cancers, and drugs targeting different steps of autophagy have been proposed as potential anticancer therapeutic approaches.

**Figure 3 biomedicines-09-01651-f003:**
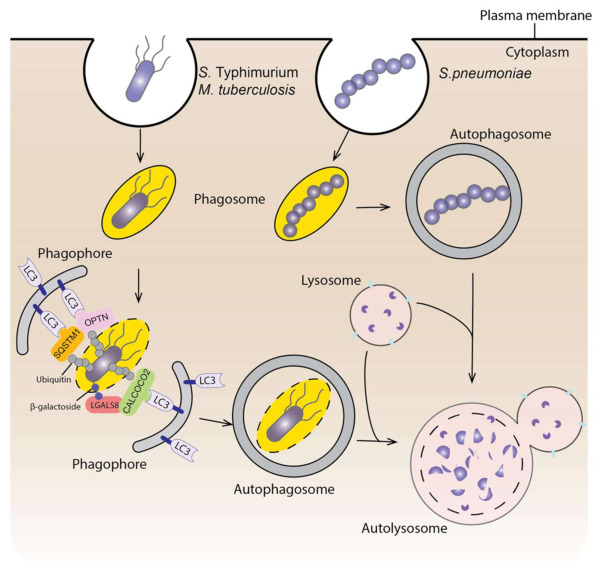
Autophagy and bacterial infection. Bacteria enter the cell within a phagosome or vacuole. Autophagy can target the bacteria within a damaged vacuole or phagosome (such as *S.* Typhimurium and *M. tuberculosis*). The ubiquitination of the bacterium and the recognition of β-galactoside by LGALS8 will recruit autophagy receptors to target the bacterium in to the phagophore. Autophagy can also target cytosolic bacteria (such as *S. pneumoniae*) causing them to be trapped within an autophagosome. Subsequently, the autophagosome will fuse with a lysosome to degrade invading bacteria.

**Figure 4 biomedicines-09-01651-f004:**
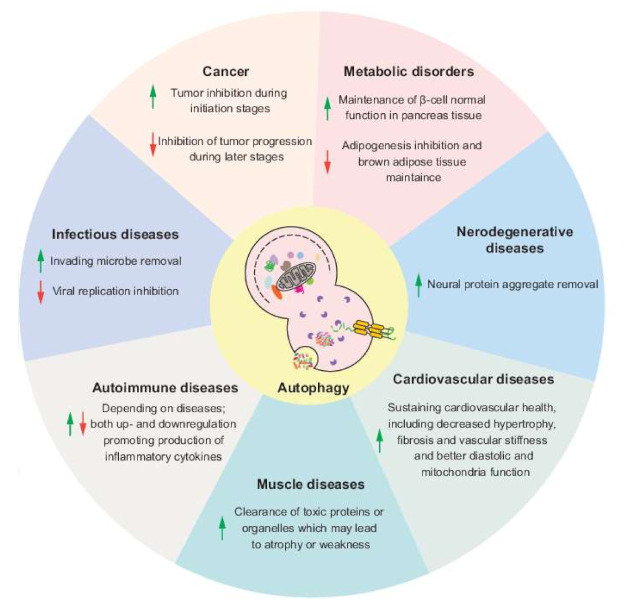
Potential effects of autophagy modulation in different human diseases. Arrows indicate the upregulation or downregulation of autophagy. Because the role of autophagy within one particular disease may be complicated and there are typically studies with controversial conclusions, we only summarize the effect reported from the majority of studies.

**Table 1 biomedicines-09-01651-t001:** PD-associated genes from GWAS and their connection with autophagy (except *SNCA*, *LRKK2*, *PINK1* and *PRKN*).

Gene Name	Description	Reference
*GBA*	Loss of GBA function impairs autophagy via PPP2/PP2A inactivation. PD-associated mutation L444P heterozygote impairs autophagy, mitochondria priming and autophagy-lysosome degradation.	[214,215]
*VPS13C*	Deletion of *VPS13C* is correlated with impaired mitochondrial morphology and upregulate PINK1-PRKN-dependent mitophagy, but the study does not show the connection between PD-associated mutations with mitophagy.	[216]
*VPS35*	VPS35^D620N^ causes autosomal-dominant Parkinson disease. VPS35^D620N^ has a reduced affinity for WASH and impairs ATG9A trafficking and localization, thus compromising autophagosome formation.	[217]
	VPS35^D620N^ impairs endosome-to-Golgi retrieval of LAMP2A and accelerates LAMP2A degradation, thus inhibiting SNCA degradation through CMA.	[218]
	VPS35^D620N^ hampers PINK1 and PRKN recruitment to mitochondria thus impairing mitophagy.	[219]
*PARK7*	PARK7 knockdown impairs autophagy and the SNCA uptake and degradation in microglia.	[220]
	PARK7 deficiency downregulates HSPA8 expression level and accelerates the degradation of LAMP2A, inhibiting SNCA degradation through CMA.	[221]
	Park7 may function in mitophagy because it is important for proper mitochondria function and Park7 upregulation can rescue the phenotype in *pink1* mutant Drosophila.	[222]
*SREBF1*	SREBF1 knockdown inhibits PRKN translocation to mitochondria, thus inhibiting mitophagy.	[223]
*FBXO7*	FBXO7^T22M^ inhibits its interaction with PRKN and impairs PRKN translocation to mitochondria.	[224]
	FBXO7^R378G^ mutation impairs ubiquitination of MFN1.	
	FBXO7^R498X^ truncation inhibits PRKN recruitment to mitochondria.	
	T22M, R378G and R498X mutations aggravate aggregation of FBXO7 in mitochondria, which may inhibit mitophagy.	[225]
*TMEM175*	TMEM175 deficiency leads to the impaired autophagosome degradation in the lysosome.	[226]
	TMEM175^M393T^ shows similar autophagosome clearance phenotype as a knockout.	[227]

**Table 2 biomedicines-09-01651-t002:** In vivo studies of the altered autophagy effect on metabolic phenotype.

Genotype	Target Organ	Modulation on Autophagy	Model	Phenotype	Reference
*Atg7^+/−^*	Whole body	Suppression	Cross with ob/ob mice	Resistance to insulin. Increased intracellular lipid content.	[371]
*sh3glb1^−/−^*	Whole body	Suppression	HFD or NCD	Higher body weight under both diets. Insulin resistance under HFD.	[372]
*Becn2^+/−^*	Whole body	Suppression	HFD or NCD	Obesity. Insulin resistance.	[373]
*atg4b^−/−^*	Whole body	Suppression	Fasting, HFD or sucrose fed	Reduced starvation-induced weight loss. Larger adipocytes in visceral fat tissue and increased hepatic steatosis. Reduced glucose tolerance and attenuated insulin sensitivity.	[374]
*Atg5 overex* *pression*	Whole body	Enhancement	NCD and aging	Lower body weight. Enhanced insulin sensitivity. Resistance to age-associated obesity.	[375]
*Becn1^F121A^* *knockin*	Whole body	Enhancement	HFD	Impaired glucose tolerance but improved insulin sensitivity.	[376]
*atg7^−/−^*	Adipose tissue	Suppression	HFD or NCD	Lower body weight. Lower white adipose tissue mass. Resistance to HFD-induced obesity. Enhanced insulin sensitivity.	[377,378]
*fto^−/−^*	Adipose tissue	Suppression	HFD or NCD	Lower white fat mass. Lower body weight.	[379]
*atg3^−/−^*	Adipose tissue	Suppression	HFD or NCD	Induced insulin resistance.	[380]
*atg16l1^−/−^*	Adipose tissue	Suppression	NCD	Induced insulin resistance.	[380]
*becn1^−/−^*	Adipose tissue	Suppression	HFD or NCD	Adipose tissue inflammation. Lower white adipose tissue mass. Lower body weight when fed with HFD.	[381]
*rubcn^−/−^*	Adipose tissue	Enhancement	NCD	Lower body weight. Lower adipose tissue mass. Glucose intolerance.	[363]
*Atg7 knockdown*	Liver	Suppression	NCD	Increased insulin resistance.	[352]
*tfeb^−/−^*	Liver	Suppression	HFD	Large and pale liver filled with lipid vacuoles. Impaired lipid degradation.	[382]
*pik3c3^−/−^*	Liver	Suppression	NCD	Lack of glycogen deposits in liver. Accumulation of lipid in liver.	[383]
*atg7^−/−^*	Pancreas	Suppression	HFD or NCD	Impaired glucose tolerance. Decreased serum insulin level.	[384]
*vma7^−/−^*	Pancreas	Suppression	HFD or NCD	Impaired insulin secretion. Glucose intolerance on HFD.	[385]
*atg7^−/−^*	Skeletal muscle	Suppression	HFD or NCD	Lower body weight on NCD. Enhanced lipid catabolism and browning of white adipose tissue when fed with HFD. Protected from HFD-induced obesity and insulin resistance.	[386]
*atg7^−/−^*	Hypothalamic POMC (proopiomelanocortin) neurons	Suppression	HFD or NCD	Higher body weight on NCD. Higher blood glucose level on HFD.	[387]
*atg7^−/−^*	Myeloid cells	Suppression	Cross with ob/ob mice	Insulin resistance. Enhanced adipose tissue inflammation.	[388]

HFD, high-fat diet; NCD, normal-chow diet.

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
