# Peer review of "The Emerging Roles of Autophagy in Human Diseases"

_biomedicines, 2021, doi:10.3390/biomedicines9111651_

Round 1

Reviewer 1 Report

Lei and Klionsky present a thorough review of the roles of autophagy in human diseases. After a brief introduction to autophagy, the authors summarize the role of autophagy in cancer, neurodegenerative diseases, infectious diseases and metabolic disorders such as obesity. They not only focus on macroautophagy, but selective autophagy as well as chaperone-mediated autophagy as well which results in an overarching understanding of autophagy in diseases. The review is well structured. In each section the review summarizes in great detail the mechanisms of autophagy associated with the disease. The authors point out the gaps of knowledge and discrepancies in the current understanding of autophagy in human diseases, highlighting the importance of further research in the field. The proposed questions after the sections are thought-provoking to the reader, however it would be nice if the authors could provide their expertise and thoughts as well as literature on trying to answer these questions. There are some cases where such answers are provided (e.g.: page 17, line 36-37), but in some cases the questions are left open (e.g.: page 7, line 26-27). Also, as in the case of autophagy in cancer therapies which are well discussed, I was missing the same discussion in the case of the other presented diseases. I would suggest adding currently known/proposed therapies based on autophagy for these as well if there are any available. I would also suggest adding one or two more figures (for example one representing the different mechanisms of autophagy at different states or cell types of cancer), further aiding the readers to understand the complex mechanisms explained in the text. 

Other comments:

In the introduction to autophagy chapter, I suggest moving the paragraph discussing CMA and microautophagy (page 3 line 18-33) forward, before discussing selective autophagy (e.g.: after page 2, line 39). This would also match the order of the subfigures of Fig. 1

After the introduction I would define which type of autophagy (I presume non-selective macroautophagy) ‘autophagy’ refers to

On page 5, line 32 the sentence “Additionally, AMBRA1, an important protein for the…” sentence is complex and hard to understand. I suggest splitting into multiple sentences for clarity.

Combine references on page 5, line 52 to [55-58]

In chapter 2.4 the phrase ‘stemness’ was not introduced earlier. Please define this as some of the readers may not be familiar with it

On page 10, line 27, the sentence “First, which autophagy step or protein…” was not clear, please rephrase

Some small typos:

Page 8, line 28 - “maintain CSCs many provide” - correct to may?

Page 10, line 12 - “which my potentially lead to” - correct to may?

Page 13, line 38 - ‘and several studies shows a decreased Aβ level” - correct to show?

Author Response

  1. “The proposed questions after the sections are thought-provoking to the reader, however it would be nice if the authors could provide their expertise and thoughts as well as literature on trying to answer these questions. There are some cases where such answers are provided (e.g.: page 17, line 36-37), but in some cases the questions are left open (e.g.: page 7, line 26-27).”

         We agree with the reviewer’s comment. We have now added additional text where appropriate (e.g., page 7, lines 29-33 and lines 49-54). However, some of these questions currently have no answers, so we have left them as open questions that will hopefully stimulate further research in these areas.

  1. “Also, as in the case of autophagy in cancer therapies which are well discussed, I was missing the same discussion in the case of the other presented diseases. I would suggest adding currently known/proposed therapies based on autophagy for these as well if there are any available.”

         Again, we think this is a good suggestion. We added information pertaining to autophagy-targeting therapies in neurodegenerative diseases (Section 3.4) and metabolic diseases (page 20, line 36 – page 21, line 16). For infectious diseases we did not focus on one particular disease other than COVID-19, for which there are currently no autophagy-based therapies, so we did not add additional information in this section.

  1. “I would also suggest adding one or two more figures (for example one representing the different mechanisms of autophagy at different states or cell types of cancer), further aiding the readers to understand the complex mechanisms explained in the text.”

         We have now added two additional figures. The new Figure 2 summarizes the different roles of autophagy in cancer, and Figure 3 show the mechanism of xenophagy; the original Figure 2 is now Figure 4. 

  1. “Other comments:

In the introduction to autophagy chapter, I suggest moving the paragraph discussing CMA and microautophagy (page 3 line 18-33) forward, before discussing selective autophagy (e.g.: after page 2, line 39). This would also match the order of the subfigures of Fig. 1.”

         We appreciate the reviewer pointing out this inconsistency. Rather than change the order of the text, we changed the order of the panels in Figure 1 to match the Introduction. Selective autophagy falls under the category of macroautophagy, so we think the current order in the text is more appropriate.

  1. “After the introduction I would define which type of autophagy (I presume non-selective macroautophagy) ‘autophagy’ refers to”

         Thank you for this suggestion. We have now added this explanation on page 1, line 31.

  1. “On page 5, line 32 the sentence “Additionally, AMBRA1, an important protein for the…” sentence is complex and hard to understand. I suggest splitting into multiple sentences for clarity.”

         We agree that the original version was not clear. We have now revised this text on page 5, lines 34-39.

  1. “Combine references on page 5, line 52 to [55-58]”

         We have now corrected this error.

  1. “In chapter 2.4 the phrase ‘stemness’ was not introduced earlier. Please define this as some of the readers may not be familiar with it”

         We agree with this suggestion. We have now added the definition on page 8, lines 28-29.

  1. “On page 10, line 27, the sentence “First, which autophagy step or protein…” was not clear, please rephrase”

         Again, we agree with the reviewer and have revised the sentence on page 10, lines 31-33.

  1. “Some small typos:

Page 8, line 28 - “maintain CSCs many provide” - correct to may?

Page 10, line 12 - “which my potentially lead to” - correct to may?

Page 13, line 38 - ‘and several studies shows a decreased Aβ level” - correct to show?”

         All of these errors have now been fixed.

Reviewer 2 Report

This review paper is well written and very helpful for readers. Autophagy is important action. T

Author Response

There were no specific comments to address.

Reviewer 3 Report

In this review, authors discuss some of the remaining questions in autophagy and diseases complex, focusing on cancer, neurodegenerative diseases, infectious diseases and metabolic disorders (focus on obesity and diabetes mellitus).

This review provides useful information on the relation between Autophagy and Diseases. However, it is too complicated and difficult to clear the relation of autophagy with the different kind of diseases, cancer, neurodegeneration, infection and metabolic disorder in one manuscript. It seems could separate into three or four different papers when you compare to the reference you cited in reference 367. Khandia R, Dadar M, Munjal A, et al. A Comprehensive Review of Autophagy and Its Various Roles in Infectious, Non-Infectious, and Lifestyle Diseases: Current Knowledge and Prospects for Disease Prevention, Novel Drug Design, and Therapy. Cells. 2019 07 03;8(7). doi: 10.3390/cells8070674). You saw that only Infectious, Non-Infectious issues have been highlighted in one review paper. To be more specific, could be more attractive and easily to be understood by general readers of Biomedicines.

Further, below have some minor questions that need to be concerned.

  1. It seems missing Table 1 (described on page 12 line 42) and Table 2 (described on page 18 line 52 and page 20 line 7).
  2. The author raise some questions in the text, however, in some of them the author did not provide the answer and suggestions. If the author could provide suggestions of those questions will be better.
  3. The author mentions the four different kinds of autophagy at the beginning, if the author could link them more tightly with the disease and provide rationales of them will be more Inspirational.
  4. Some minor typo error needs to be fixed, such as

on page 5 line 52 'in more benign diseases rather than invasive cancer [55] [56, 57] [58], indicating that....', 

on page 7 line 5 "energy starvation, RB1C,C1 is activated by ULK1 and inhibits focal adhesion kinase... "

on page 9 line 50, "gemcitabine in pancreatic ductal adenocarcinoma [147], Of note, autophagy-independent effects may occur when treating cancer cells with CQ, but these effects...."

Author Response

  1. “This review provides useful information on the relation between Autophagy and Diseases. However, it is too complicated and difficult to clear the relation of autophagy with the different kind of diseases, cancer, neurodegeneration, infection and metabolic disorder in one manuscript. It seems could separate into three or four different papers when you compare to the reference you cited in reference 367. Khandia R, Dadar M, Munjal A, et al. A Comprehensive Review of Autophagy and Its Various Roles in Infectious, Non-Infectious, and Lifestyle Diseases: Current Knowledge and Prospects for Disease Prevention, Novel Drug Design, and Therapy. Cells. 2019 07 03;8(7). doi: 10.3390/cells8070674). You saw that only Infectious, Non-Infectious issues have been highlighted in one review paper. To be more specific, could be more attractive and easily to be understood by general readers of Biomedicines.”

         We certainly understand the reviewer’s point; however, our goal in this review was to demonstrate the close connection between autophagy and a range of diseases. In addition, we wanted to provide a single reference that would give readers an overview of this topic. Along these lines, we chose not to go too deeply into mechanism, but rather to summarize the relevant information about how modulation of autophagy affects the disease state. Nonetheless, to help make the review easier to follow, and in line with comments from Reviewer #1, we have added two more figures.

  1. “It seems missing Table 1 (described on page 12 line 42) and Table 2 (described on page 18 line 52 and page 20 line 7).”

         We apologize for this oversight; the tables were initially prepared as separate documents, but we have now incorporated them directly into the manuscript.

  1. “The author raise some questions in the text, however, in some of them the author did not provide the answer and suggestions. If the author could provide suggestions of those questions will be better.”

         This is essentially the same point made by Reviewer #1, comment #1. As above, we agree with the reviewer’s comment. We have now added additional text where appropriate (e.g., page 7, lines 29-33 and lines 49-54). However, some of these questions currently have no answers, so we have left them as open questions that will hopefully stimulate further research in these areas.

  1. “The author mentions the four different kinds of autophagy at the beginning, if the author could link them more tightly with the disease and provide rationales of them will be more Inspirational.”

         This is a good suggestion. We have now added a paragraph at the end of the Introduction (page 3, lines 20-32) indicating the importance of autophagy under stress and normal conditions. Because autophagy is important for both situations, dysregulation of autophagy may lead to a diseased state due to changes in these critical physiological processes.

  1. “Some minor typo error needs to be fixed, such as

on page 5 line 52 'in more benign diseases rather than invasive cancer [55] [56, 57] [58], indicating that....', 

on page 7 line 5 "energy starvation, RB1C,C1 is activated by ULK1 and inhibits focal adhesion kinase... "

on page 9 line 50, "gemcitabine in pancreatic ductal adenocarcinoma [147], Of note, autophagy-independent effects may occur when treating cancer cells with CQ, but these effects...."”

         We appreciate the reviewer pointing out these errors, which have all been fixed.

Round 2

Reviewer 3 Report

All suggestions and comments were addressed by the authors. I have no
further comments or suggestions